# A Simulated Organic Vegetable Production and Marketing Environment by Using Ethereum

**Dong-Her Shih [1],\* , Kuan-Chu Lu [1], Yi-Ting Shih [1] and Po-Yuan Shih [2]**

[1]  Department of Information Management, National Yunlin University of Science and Technology, 123, Section 3, University Road, Douliu, Yunlin 64002, Taiwan; qq1214x@yahoo.com.tw (K.-C.L.); elaine15999@gmail.com (Y.-T.S.)

[2]  Department of Finance, National Yunlin University of Science and Technology, 123, Section 3, University Road, Douliu, Yunlin 64002, Taiwan; D10424003@gemail.yuntech.edu.tw

\*   Correspondence: shihdh@yuntech.edu.tw; Tel.: +(886)5-534-2601 (ext. 5340)

**Abstract:** Agriculture is indispensable for every country. Farmers use pesticides in large quantities to increase productivity, thus causing serious damage to the agro-ecological environment. If consumers consume a large number of such vegetables, there can be an adverse effect on their health. Therefore, consumers shift from consuming general vegetables to organic vegetables. However, the price of organic vegetables is higher than that of general vegetables, and there are doubts about the authenticity of the production and marketing process. Unfortunately, production and sales records are often falsified. The blockchain may be able to guarantee the authenticity of organic vegetables because the blockchain is tamperproof and transparent. Therefore, we proposed an exercise environment for the production and marketing of organic vegetables by using Ethereum. This proposed system can ensure the authenticity of the production and sales record, it may increase the sales of organic vegetables and solve the problem of agricultural ecological environment pollution in the real world.

**Keywords:** organic vegetables; TAP (Traceability Agricultural Product); blockchain; Ethereum

## 1. Introduction

In recent years, people have paid an increasing amount of attention to quality of life challenges, and quality of life has become the most crucial topic in modern society. In Taiwan, food safety problems cause people to panic every year. For example, common processed food products contain many chemicals, which causes health problems. Moreover, agricultural products that people directly consume are sometimes unprocessed. As the agro-ecological external environment (such as water resources) can be polluted during the process of agriculture, the resulting unprocessed agricultural products are polluted. People who consume these products for a long time due to ignorance exhibit negative effects on their own health and the health of future generations, such as chronic diseases and fetal growth retardation [1]. Many farmers use inappropriate toxic pesticides in vegetables before and after harvesting, thus causing health problems for farmers and consumers and environmental pollution [2]. Food safety is a common problem for people all over the world. On New Year's Eve, 2014, thousands of people were poisoned in Japan, because their food was contaminated with organophosphorus pesticides.

The food safety problem is constantly emerging, and people's requirements of the quality of agricultural products are increasing with the quality of life. Therefore, people pay more attention to organic, healthy, and safe agricultural products [3]. The results of a study by Dipeolu et al. [4] indicate that people believe that organic vegetables are healthier than traditional vegetables and exhibit no harmful effects. Consumer demand for organic food is growing, and the most crucial motivation

for opting for organic foods is health problems [5]. Su Murong, the CEO of the Organic Verification Agency, Taiwan, said that in addition to on-site and pesticide residue testing, the most crucial thing is the farmer's planting record. We cannot guarantee that the organic vegetables we buy do not contain agricultural residues and that they are really produced in an organic environment. In the market, fake organic products can be seen everywhere. The available standards are not static, and the authenticity of organic vegetables cannot be confirmed. To solve these problems, blockchain technology can be used for credit certification of organic agricultural products; blockchain can deliver features such as Trust, Openness, Collective Maintenance, and Tamper proofing [3]. Organic agricultural products lack a proper retail space, continuous supply, and certification, and many inferior products are distributed [6]. Consumers have various expectations regarding product attributes, such as price, quality, and certification [7]. Providing superior organic agricultural products at reasonable prices can not only increase consumer purchases but also protect the global ecological environment [5]. Therefore, the blockchain technology was used in the production and sales stages for organic vegetables. Due to the open, transparent, and non-tamperable characteristics of the blockchain, it can be guaranteed that farmers have grown vegetables according to the standards of organic vegetables. In the organic vegetable verification stage, a stamp is given after verification. As the blockchain cannot be falsified, the batch of organic vegetables verified is genuine. In the sales stage, farmers sell the organic vegetables at low prices, and the middlemen sell those vegetables at high prices, thus resulting in high profits for the middlemen. Therefore, when organic vegetables are sold to the consumers, the prices could be unreasonable. Due to the blockchain technology and its characteristics of openness and transparency, the middlemen cannot deliberately increase prices. This solves the problem of ecological environment damage caused by the long-term use of pesticides, health problems, and the unreasonable price of organic vegetables.

The research topics are as follows:

- The record of organic vegetable production and sales cannot prove authenticity.
- The prices of organic vegetables are increased.
- Agricultural ecological environment pollution problems are caused.
- The purposes of this study are as follows:
- To propose an organic vegetable production and marketing system based on Ethereum.
- To integrate and analyze the system on the Ethereum platform.

Section 2 introduces the blockchain technology. Section 3 presents the proposed blockchain Ethereum solution in detail. Section 4 describes the system implementation details. Section 5 provides the test details. Section 6 presents the conclusion.

## 2. Blockchain Technology

A typical blockchain system is composed of a set of blocks that are linked together to form a chain. The system can concatenate and protect serialized transaction records through cryptography. Each block contains a cryptographic hash function, a timestamp, and the transaction data of the previous block. Due to this design, the contents of the block cannot be tampered. The blockchain is a decentralized ledger that links each block into a chain without the need for a third party, but instead generates a consensus in the form of decentralization in the open network center. After ensuring the credibility of the blockchain, that is, once a new consensus is attained, all participants update their ledgers simultaneously. The main components and characteristics of the blockchain are described as follows:

- Peer-to-peer (P2P): in a peer-to-peer network, all nodes communicate with each other. This network deals with other nodes and can be decentralized without using third party software. In other words, the status of each node in the network is the same. No node belongs to the central control position, and no other node needs to serve an intermediary role in the transaction. Nodes

in a blockchain network can join or exit based on their will, and nodes can perform most of the functions. If a node is damaged, added, or removed, the operation of entire system is not affected. Instead, the robustness of the network increases with the number of nodes.

- Participants in the blockchain network: two types of participants exist in the blockchain network. The first type of participant is a user and serves the functions of money transfer (e.g., Bitcoin and Ether) and execution currency remittance. The second type of participant is a miner. A miner functions as a production block and gets paid.

- Transaction: this command is issued by the user while making money remittances and is transmitted through each node in the blockchain network. A transaction includes the following stages in its lifecycle: establishing a transaction, naming it, passing it, and saving it in a block. First, the transaction is established, and then, the transaction command is passed to each node. After receiving the transaction node, the system checks whether the transaction meets certain conditions. If the conditions are met, a success message is sent back to the transaction source node, and the transaction command is passed to each node. If the conditions are not met, the discarded message is returned, and the transaction command is not passed to other nodes. The final transaction is verified by the miner and recorded on the blockchain, and the block is passed to each node, which is recorded on a decentralized ledger.

- Mining: the production of a new block is known as mining. If mining is successful, additional money is generated, and the miner earns the currency as a handling fee. There can be only one miner in the new production block, and only that miner can earn the handling fee. To earn the handling fee, many miners compete for obtaining new blocks. However, the production of new blocks requires high computing power. Therefore, in mining, there is a saying that "miners compete with each other to produce blocks."

- Decentralized ledger: in the blockchain network, all transferred blocks are stored on the node's ledger. The information of the preceding block is stored in each block. By referring to the preceding block through the hash value, the blocks are connected to each other in a chain format. Each node stores the record of all transactions, and the contents of all ledgers are exactly the same, which represents the consensus mechanism of the blockchain. All the transaction information in the decentralized ledger is encrypted, transparent, non-tamperable, and reliable. If a ledger is damaged, the record can be viewed on other nodes; hence, there is no downtime. Decentralized ledgers can be applied to financial insurance, medical health, government and public welfare, copyright and ownership, academic and human resource management, shared economies, and Internet of Things.

- Encryption hash: a hash function is an encryption technique in the blockchain network. Common hash algorithms are SHA-256, RIPEMD-160, and HASH160. A characteristic of a hash function is that the input string has an arbitrary size, but the size of the output string is a fixed. For an input value, a completely unrelated output value known as the hash value is obtained. By using collision resistance, two different outputs must be produced for two different inputs. The function also has hidden or irreversible characteristics. Moreover, it is highly difficult to trace back the original input from the hash values. The hash algorithm can be used to verify whether the message has been tampered. Moreover, the verification can be simplified using a Merkle tree. A Merkle tree repeats the hashing process of thousands of transactions by forming new hash values in pairs and finally produces a final set of hash values. Hence, the Merkle tree can substantially reduce the amount of resource consumption during transmission and computation. A blockchain mainly uses the elliptic curve digital signature algorithm. The node user in each blockchain possesses both public and private keys. The public key is known to other users in the blockchain network. The node user must know the private key to receive currency, sign electronically, and send money. This key should not be disclosed to the others in the network. While trading, the preceding transaction and payee's public key must be hashed to create a digital signature, which is added to the back of an electronic money digital signature.

- Timestamp: a timestamp is used to ensure that the block sequence is written on blocks. Each block is hashed and then a timestamp is added and passed. This timestamp is used to prove the validity of data at a specific time. Each timestamp is hashed with the previous timestamp, which is then hashed with the next timestamp. Thus, a chain that ensures a sequence of blocks is formed. If a timestamp is generated for a block, the block cannot be modified. If the block hashed value does not match, the decentralized ledger immediately detects that tampering has occurred.

The currency circulating on the blockchain network is known as "virtual currency or cryptocurrency." For example, a Bitcoin network generates Bitcoins and Ethereum generates Ether. Although both Bitcoin and Ethereum networks use the blockchain technology, both networks are separate technologies. In 2008, Satoshi Nakamoto proposed the concept of "blockchain" in Bitcoin White Paper, implemented the Bitcoin network in 2009, and developed it on GitHub in an open source software [8]. The Bitcoin network is open to all in terms of participation. If a network is open, it is called "Public." A network that needs permission for participation can be called "Private" or "Consortium." A Bitcoin is a decentralized ledger with a hash algorithm that forms a transaction record of a cryptographic electronic currency that cannot be tampered.

Among the consensus mechanisms of blockchains that use Proof of Work, Bitcoin is the most representative, and its consensus protocol mainly comprises workload proof and the longest chain mechanism. In a Bitcoin blockchain, all the transaction messages over a period of time are packaged into one block and added to the longest blockchain, thus verifying that the node of the successful new blockchain will reward the successful verifier. The whole process of packing the ledger and obtaining the reward is known as mining, and the node participating in the block competition is called a miner.

## 3. Simulated System of Organic Vegetable Production and Marketing

In this section, we introduce the Ethereum blockchain technology solution to organic vegetable production and sales from a case study. Then, we introduce all the roles in the simulated system of organic vegetables production and marketing. Next, we present how each role performs in a smart contract.

### 3.1. Case Study from Albertsons Companies

Albertsons Companies (Eugene, Oregon, USA), which operates nearly 2300 stores across America is one of the largest food and drug retailers in the United States. Food safety is a very significant step in Albertsons. Through the provenance of the products, the ability to track every move from the farm to the customer's basket, can be very empowering for their customers. Albertsons plans to pilot IBM (International Business Machines Corporation) Food Trust to help overcome the obstacles that have existed when a traceback is initiated for high-risk food products like romaine lettuce. The leading grocery company will start by tracing bulk romaine lettuce from one of its distribution centers and will then explore expanding to other food categories throughout its distribution network. The need to find more efficient ways of tracing products and identifying likely sources of contamination in a timely manner. Consequently, retailers are exploring new technologies to improve the infrastructure that underpins the global food supply chain. Moreover, Albertsons will help bring food traceability to both consumers and industry players.

Moreover, a system of record that can be used to trace and authenticate objects as they move through the supply chain, a digital record of every transaction or interaction can be created for food products. From a packaging date, to the temperature at which an item was shipped, to its arrival on a grocery store shelf, this increased transparency can address a broad range of food quality issues for major retailers and consumers.

*3.2. Simulated System Overview*

Base on the case study from Section 3.1, all the roles served in the simulated system, namely farmer, consumer, intermediary, retailer, organic inspection agency, arbitrator, and Ethereum smart contract is shown in Figure 1. The yellow line indicates the actual transaction. Farmers can sell organic vegetables to consumers through middlemen and retailers, or farmers can directly sell them to consumers. The gray lines indicate that each role interacts with the Ethereum smart contract. All roles are summarized as follows:

- Farmer: the functions of a farmer include purchase, registration, production, sales, and updating. In the purchase function, farmers buy organic fertilizers and seeds from fertilizer traders and seed traders. In the registration function, farmers are responsible for registering the organic vegetable production and sales resume information into the Ethereum network. Farmers are responsible for all processes of producing organic vegetables. In the sales function, farmers sell organic vegetables to consumers or middlemen. In the updating function, farmers are allowed to update the organic vegetable production and sales resume information pertaining to the seed information, fertilizer information, fertilization records, organic vegetable sale prices, and update the ownership details of the organic vegetables.

- Intermediary: the functions include purchase, sale, and updating. In the purchase function, organic vegetables are bought from farmers. In the selling function, these vegetables are sold to retailers. In the updating function, the middlemen update the organic vegetable production and sales resume information in terms of the price at which the organic vegetables are sold and update the ownership details of the organic vegetables.

- Retailer: the functions include purchase, sale, and updating. In the purchase function, organic vegetables are purchased from middlemen. In the sales function, the vegetables are sold to consumers. In the updating function, retailers update the organic vegetable production and sales resume information in terms of the price at which the organic vegetables are sold and update the ownership details of the organic vegetables.

- Consumer: the functions include purchase and updating. In the purchase function, organic vegetables are purchased from farmers or retailers. In the update function, consumers update the ownership details of the organic vegetables. This study defines that consumers are not the ultimate organic vegetable owners. Therefore, it allows first-hand purchasing or even auctions. Products are still resold to other consumers after purchase, so consumers can update the price or ownership of organic vegetables.

- Organic inspection agency (OIA): the functions include soil testing, water quality testing, organic vegetable testing, and adding functions. The soil and irrigation water in the environment in which organic vegetables are produced must pass organic standards. An OIA should be involved to conduct on-site inspections. Before selling organic vegetables, the vegetables must first be sent to an OIA for organic testing. The new feature allows organic inspection agencies to add inspection results to the organic vegetable production and sales resume information.

- Arbitrator: when all transactions are in dispute, the system requires an arbitrator to transfer the controversial Ether to the arbitrator. Subsequently, the arbitrator assesses the right and wrong parties in the dispute. The dispute may be because the payment is completed and the consumer did not receive the organic vegetables.

- Ethereum smart contract: in this system, the smart contract is divided into Traceability Agricultural Product contract (TAP contract), organic vegetable contract, and manage contract.

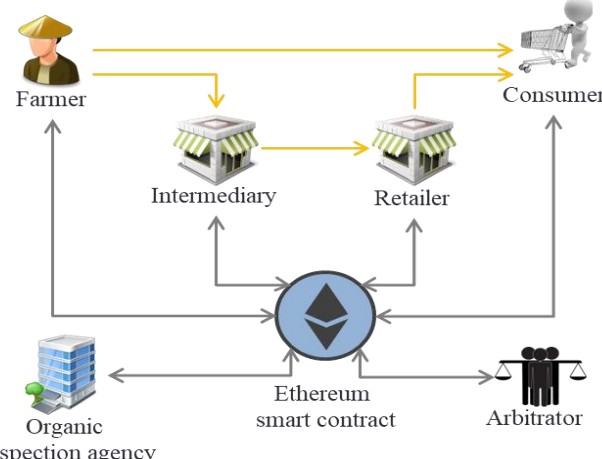

**Figure 1.** All roles and the Ethereum smart contract of the simulated system.

A case scenario of real-life transactions for production, sales, and dispute are shown in Figure 2. First, the soil in the plantation and the irrigation water used for planting must be applied for inspection and regular tracking to the Organic Inspection Agency (QIA). Farmers will record the organic vegetable production and sales resume (TAP) through the inspection proved mark. Farmers must also record seed information, seed dealers, and transaction to the TAP of organic vegetables. Fertilizer information, fertilizer dealer, transaction, and fertilization details must also be recorded in the TAP. The entire production process can be introduced into the Internet of Things technology, and the organic vegetable TAP is built on the Internet of Things system. Farmers can set up temperature, humidity, pH, and other sensors and global status systems (GPS) in organic vegetable TAP for big data analysis. When the vegetables are harvested, the producers must send the vegetables to an OIA for inspection. OIA will give organic inspection labels. Through the label, consumers can trace the production and sales resume, product name, tracking code, verification institution name, and disclosure information.

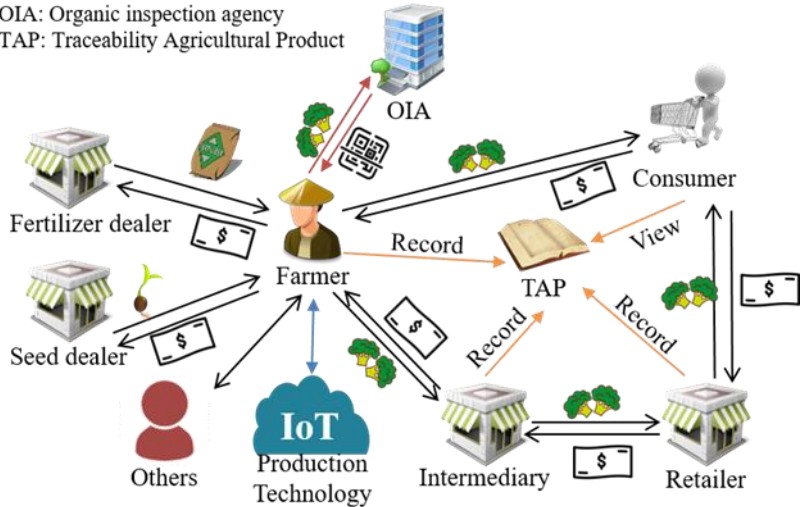

**Figure 2.** A simulated scenario of production and marketing of organic vegetables.

The sales phase can be divided into two pathways. A pathway for farmers can be produced and sold by himself. Another pathway is farmers sell organic vegetables to intermediaries, intermediaries sell to retailers, and consumers buy organic vegetables from retailers. Information about intermediaries and retailers must also be recorded in the TAP.

*3.3. Smart Contract Design*

In this section, we explain how each role operates and is available in the TAP contract, organic vegetable contract, and manage contract. A TAP contract serves to record the entire production process. In this contract, the results of the inspection of organic vegetables are noted; the contract for the transaction is generated by the organic vegetables contract; the content of that contract is current ownership of the TAP contract site, ownership of the organic vegetables, organic Vegetable information, and transaction times. The Manage contract enables organic vegetable sellers to manage all organic vegetables contracts in a unified manner. Users of this system can view the current trading status. One possible status is "Not yet traded, not received, and the transaction is completed."

The TAP contract is an organic vegetable production and sales resume contract, as shown in Figure 3. A farmer must first be registered, and all production processes should be updated into this contract. Before the sale of organic vegetables, the products are sent to an OIA to verify whether they meet the organic standards. Therefore, the inspection results are added by the OIA. These details can be only added once, and no modification is allowed. All production processes are made available to the public, and organic inspection results can only be used once to solve consumers' doubts about the authenticity of the organic vegetable production process and inspection results.

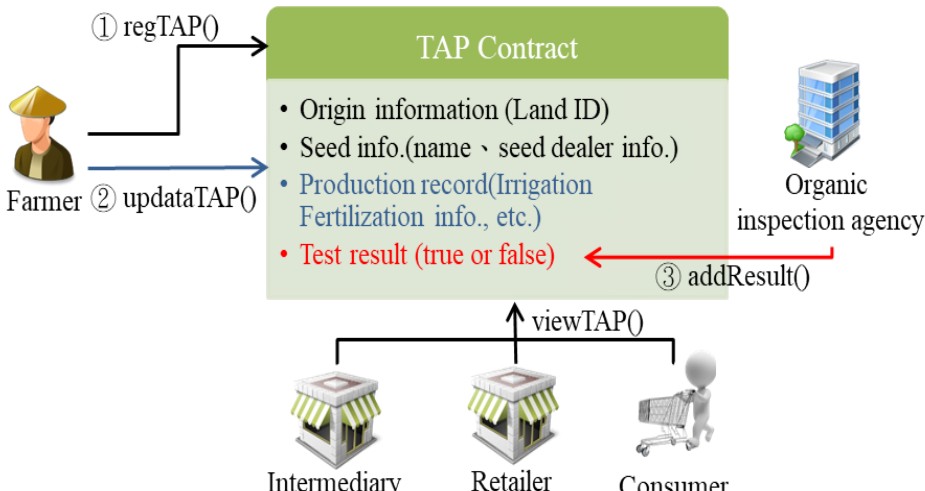

**Figure 3.** Function and accessibility of every role in the Traceability Agricultural Product (TAP) contract.

Figure 4 displays how farmers trade in this system. To facilitate the management of each organic vegetable contract, the organic vegetable contract status is added to the management contract, and the initial value of the status is "not yet traded." When a consumer wants to buy the organic vegetables, the activate() function is activated to make the transaction and enable payment using Ether. When activate() is initiated, it automatically judges whether the organic vegetables are expired. If the products are expired and the payment is completed, the status in the management contract will be revised to "not received." The consumers do not find the price unreasonable because the blockchain is open and transparent, thus preventing profiteers from increasing prices at their will. A farmer uses the withdraw() function to collect the Ether. After the payment is successful, the transaction is completed. At this point, the ownership of the organic vegetable contract will change, so the ownership of the organic vegetable contract will be automatically changed, and the status of the management contract will be changed to "transaction completed."

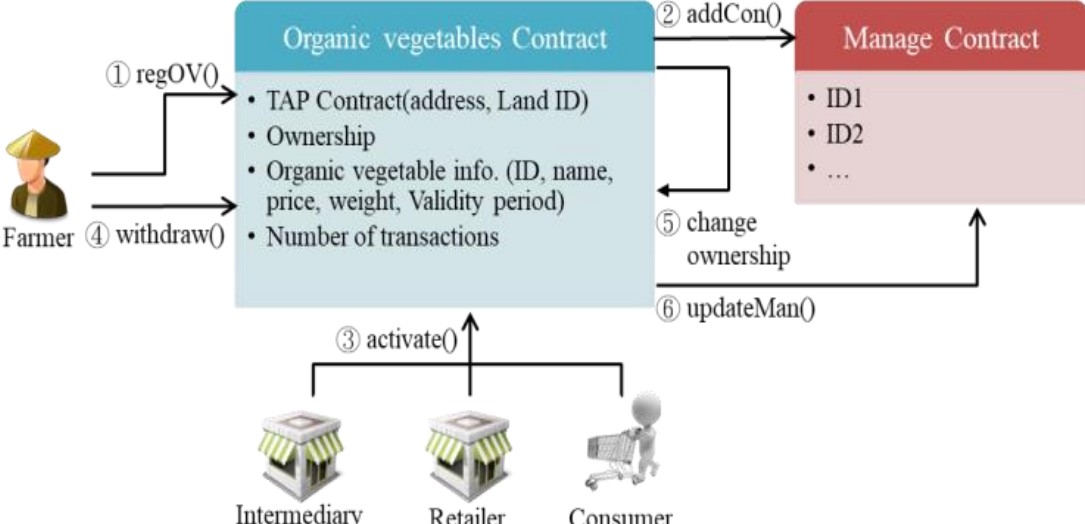

**Figure 4.** Function and accessibility of every role in the organic vegetable contract (ownership for farmer).

When the organic vegetable contract ownership is changed to intermediary or retailer while they conduct organic vegetable trading, they do not need to register an organic vegetable contract. However, all the other processes are the same as those for farmers, as shown in Figure 5. The figure provides an example when a retailer holds the ownership of vegetables.

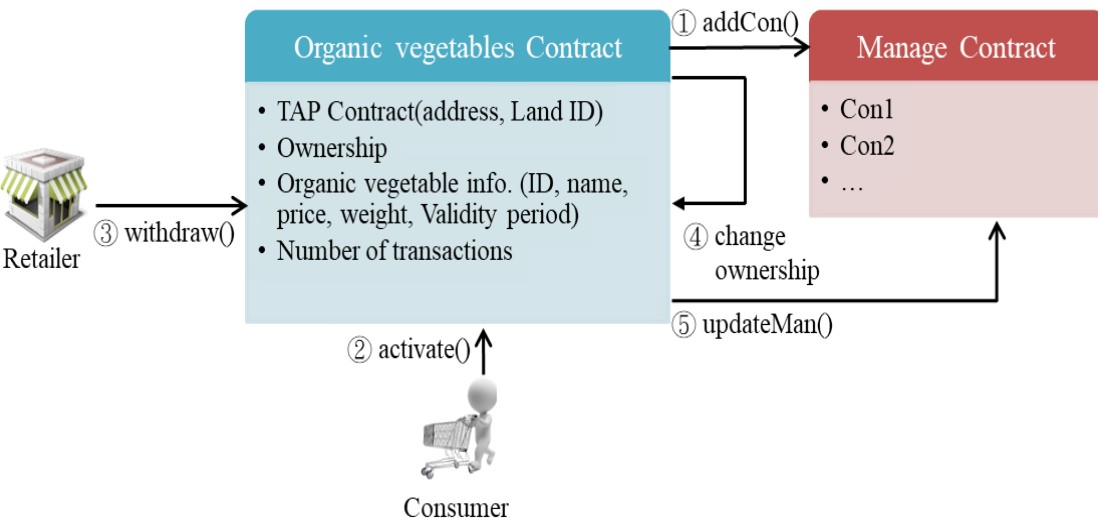

**Figure 5.** Organic vegetable contract, ownership is retailer.

## 4. Implementation

In terms of blockchain technology, Ethereum's resources are easy to obtain, and it is an open source. Therefore, this study was carried out using Ethereum under the cost of the public. Blockchain's research by using Ethereum are covered by many areas [9–13]. All the blockchain functions in Section 3 and algorithms used in this section are integrated into a flowchart as shown in Figure 6 for clarity.

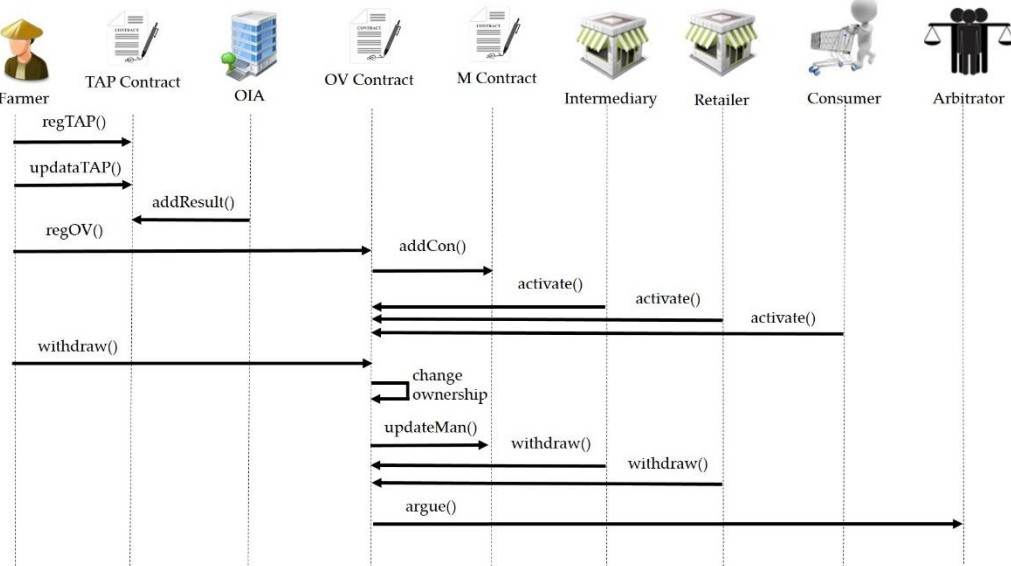

**Figure 6.** Flow chart of blockchain function and algorithm used in organic vegetable TAP.

The programming language for writing smart contracts in this study is known as Solidity. Solidity is a contract-oriented programming language that was designated by Ethereum. All contracts were created (developed) and tested using the Remix Integrated Development Environment (IDE), which is a Solidity official online IDE that provides testing and debugging capabilities so it can be easily compiled. This section describes the algorithms that implement the system.

### 4.1. Registered and Update Tap

Algorithm 1 displays the process for registering the production and sales history. A TAP contract is used to register the production and sales resume information. First, the registered production and sales resume must prove that the user is a farmer. After that, it will be checked whether the farmland code has been repeated. If not, the registration action can be performed.

---

**Algorithm 1:** Registered TAP

---

    **Input:** msg.sender, farm_num
**Output:** str
**1 if** *msg.sender = Farmer* **then**
**2**     **if** *farm_num = 0* **then**
**3**       register TAP information(farm_num, Seed info.) storage in the TAP Contract.
**4**       str = regTAP success!!!
**5**     **else**
**6**       str = farm_num can't repeat!!!
**7**     **end**
**8 end**
**9** return str

---

Algorithm 2 presents the production record in the production and sales record. When the update production record function is initiated, the permission judgment is first performed. Only the person who registered in the contract can update the organic vegetable production record. This function prevents a malicious person from falsifying the production record at will.

---

**Algorithm 2:** Update TAP

---

    **Input:** msg.sender, farm_num
**Output:** str
**1 if** *msg.sender = Farmer* **then**
**2**    **if** *farm_num = 0* **then**
**3**        str = farm_num no register!!!
**4**    **else**
**5**        input production record to TAP(Fertilization, Irrigation, etc.) storage in the TAP Contract.
**6**        str = updataTAP success!!!
**7**    **end**
**8 else**
**9**    str = Permission denied!!!
**10 end**
**11** return str

---

### 4.2. OIA Addition Result

New inspection results can only be made if the production and sales history has been registered to the TAP contract and there is a production record. Algorithm 3 presents the details of the new test results. The first call to the new test result function can only be an OIA, so the permission check is performed. In the second step, it will be confirmed whether the test result can be added. This step ensures that this action can only be performed once. The type of test result is stored. The system stores the information in Boolean data type; thus, the value of the input result can only be "True" or "False." True represents passed, and False is failed.

---

**Algorithm 3:** Organic Inspection Agency add result

---

    **Input:** msg.sender, farm_num, state, record
**Output:** str
**1 if** *farm_num = 0* **then**
**2**    str = farm_num no register!!!
**3 else**
**4**    **if** *msg.sender = OrganicInspectionAgency* **then**
**5**      **if** *state = false* **then**
**6**        **if** *record > 0* **then**
**7**          input organic vegetables inspection result storage in the TAP Contract.
**8**          state = true
**9**          str = addResult success!!!
**10**        **else**
**11**          str = No production record!!!
**12**        **end**
**13**      **else**
**14**        str = test results have been completed!!!
**15**      **end**
**16**    **else**
**17**      str = Permission denied!!!
**18**    **end**
**19 end**
**20** return str

---

### 4.3. Organic Vegetable Transaction

This process involves the trading phase of organic vegetables. We have three major steps, namely, registration of organic vegetables, payment, and collection. First, only the farmer has the right to

register organic vegetables, and registration can be completed for subsequent transactions. Algorithm 4 presents the method by which the farmer can register. After the registration is successful, the organic vegetable contract transaction status is added to the manage contract.

---

**Algorithm 4:** Organic vegetables Transaction(register)

---

    **Input:** msg.sender, ov_ID, state
**Output:** str
**1 if** *msg.sender = Farmer* **then**
**2**    **if** *ov_ID = 0* **then**
**3**       register organic vegetables information(weight, price, etc.) storage in the Organic Vegetables Contract and Manage Contract.
**4**       state = true; str = regOV success!!!
**5**    **else**
**6**       str = ov_ID can't repeat!!!
**7**    **end**
**8 else**
**9**    str = Permission denied!!!
**10 end**
**11 return** str

---

Algorithm 5 presents how consumers can make payments. Consumers can be divided into general consumers, middlemen, and retailers. We use the payable method in Solidity (an object-oriented, high-level language for implementing smart contracts) to transfer Ether to an organic vegetable contract. When the consumer makes a payment, the system first determines whether the organic vegetable is in the controversial stage. Moreover, it is impossible to trade if the transaction is in the dispute stage. When the payment is successful, the status of the organic vegetable contract in the management contract is updated.

---

**Algorithm 5:** Organic vegetables Transaction (payable)

---

    **Input:** msg.sender, ov_ID, msg.value, argue, state
**Output:** str
**1 if** *msg.sender = Owner* **then**
**2**    str = activate failure!!!
**3 else**
**4**    **if** *ov_ID = 0* **then**
**5**       str = ov_ID no register
**6**    **else**
**7**       **if** *argue > 0* **then**
**8**         str = arguing . . .
**9**       **else**
**10**         **if** *state = true* **then**
**11**           Transfer msg.value to the Organic Vegetables Contract and update Manage Contract state.
**12**           state = false
**13**           str = activate success!!!
**14**         **else**
**15**           str = activate failure!!!
**16**         **end**
**17**       **end**
**18**    **end**
**19 end**
**20 return** str

---

The final step is to collect the money. Algorithm 6 presents the method by which the organic vegetable owner collects money. The Ether in the organic vegetable contract is remitted to the owner's wallet. After the payment is successful, the organic vegetable contract transaction status is added to the management contract, and the organic vegetable owner is revised to the new purchaser.

---

**Algorithm 6:** Organic vegetables Transaction(withdraw)

---

    **Input:** msg.sender, state
**Output:** str
**1 if** *msg.sender = Owner* **then**
**2**    Transfer the Organic Vegetables Contract balance to Owner's wallet, update Manage Contract state and change organic vegetables ownership.
**3**    state = true
**4**    str = withdraw success!!!
**5 else**
**6**    str = Permission denied!!!
**7 end**
**8** return str

---

*4.4. Dispute Resolution*

There may be disputes during a transaction. If the farmer sells the organic vegetables to the consumer and does not receive the payment, the farmer will not be able to update the status of the smart contract, which will result in a dispute. At this point, the transaction will be stopped first and Arbitrators are arbitrated [14]. Algorithm 7 shows how disputes are handled in this system. When a dispute arises, the dispute information is stored in the organic vegetable contract, and the status of the organic vegetables in the management contract is changed. If there is a balance in the organic vegetable contract, the balance is transferred to the cultivator, and the organic vegetable is considered unable to be traded. The subsequent processing is handed over to the cultivator to judge the dispute.

---

**Algorithm 7:** Dispute resolution

---

    **Input:** msg.sender, ov_ID, reason, state
**Output:** str
1 if *ov_ID = 0* **then**
**2**    str = No ov record
**3 else**
**4**    Input dispute reason, argue caller information to Organic Vegetables Contract and update Manage Contract state.
**5**    Transfer the Organic Vegetables Contract balance to Arbitrator and update Manage Contract state.
**6**    state = false
**7**    str = argue success!!!
**8 end**
**9** return str

---

## 5. Testing and Validation

This study currently focuses on the technology used in the production and marketing of organic vegetables with blockchain, so all testing and validation is only for the blockchain function. The system works in reality will be studied in the future to establish a complete system and participate in role interaction. However, the source code of Ethereum in this study can be found at https://github.com/elaine15999/solidity for demonstration. This section describes the details of using the Remix IDE to test the smart contract code and perform cost and security analyses. Table 1 presents the addresses of

all accounts in this system. All accounts are named as Farmer, Intermediary, Retailer, Consumer, OIA, and Arbitrator. All of the following tests were conducted using these six accounts.

**Table 1.** All account addresses.

| Account | ADDRESS |
|---------|---------|
| Farmer | 0xca35b7d915458ef540ade6068dfe2f44e8fa733c |
| Intermediary | 0x14723a09acff6d2a60dcdf7aa4aff308fddc160c |
| Consumer | 0x4b0897b0513fdc7c541b6d9d7e929c4e5364d2db |
| Retailer | 0x583031d1113ad414f02576bd6afabfb302140225 |
| OIA | 0xdd870fa1b7c4700f2bd7f44238821c26f7392148 |
| Arbitrator | 0xdbc3a811ec4a57b370892cc11790d9e3d65d41cb |

*5.1. Register and View TAP*

Figure 7 presents the details of the production and sales history contract. The land code is entered in the register to view all the information about this production and sales record. The address specified as Index 0 is the address of the Farmer and indicates the ownership of the registered production and sales record. The following register stores the land code, seed name, seed supplier, seed supplier, OIA address, registration time, total number of production records, and whether the test results can be added. To see the production record information, the land code is entered and the first data are available in view TAP. Then, the production record information can be viewed.

*5.2. Organic Vegetable Transaction*

Before selling this batch of organic vegetables, the farmer must register in the organic vegetable contract. After registering in viewOC(), the organic vegetable number should be entered and the details of the organic vegetables should be viewed. The price of organic vegetables, the ownership of organic vegetables, the number of transactions, the name of organic vegetables, and the weight of organic vegetables are displayed in sequence, as shown in Figure 8. The address of the organic vegetable ownership presents the location of the farmer, and the number of transactions is also shown as zero. This organic vegetable is registered for the farmer. Within a short time, some consumers want to buy this organic vegetable, as shown in Figure 9. The organic vegetable number that has to be purchased in activate() is entered, and transact if pressed to perform the remittance action. The management contract message "Uncollected" is displayed after the remittance is completed. The farmer then collects the money. This is because the current ownership of the organic vegetables belongs to the farmer, so only the farmer can collect the money. After the payment and transaction are completed, the ownership of the organic vegetables is changed. As shown in Figure 10, the ownership address is changed to the address of the consumer, and the number of transactions is increased by one.

A detailed step of transactions scenario is shown in Figure 11. First, the farmer issues a transaction. Second, the transaction is transmitted to all nodes in this blockchain. Third, it is assumed that Miner A has won the mining and posts the transaction block 10 in this Ethereum network. Finally, block 10 is transmitted to all nodes and recorded in the decentralized ledger.

register

: 2745

call

0: address: reguseraddr
0xCA35b7d915458EF540aDe6068dFe2F44E8fa733c
1: uint256: farm_num 2745
2: string: s_name  Organic Spinach Seedlings
3: string: sd_name  Organic company
4: uint256: sd_UBN 2165532
5: address: OIAaddr
0xdD870fA1b7C4700F2BD7f44238821C26f7392148
6: uint256: nowtime 1553869661
7: uint256: num 2
8: bool: resstate false
9: bool: res false

str

viewTAP

_farm_num: 2745

num: 1

call

0: string: Record description
1: string: Fertilization
2: string: Company name
3: string: Organic Fertilizer  Company
4: string: Uniform  Company Number
5: uint256: 3665514
6: string: Time
7: uint256: 1553869757

**Figure 7.** Register and TAP view in proposed system.

viewOC

_ov_ID: 23-1

call

0: uint256: 4
1: address:
0xCA35b7d915458EF540aDe6068dFe2F44E8fa733c
2: uint256: 0
3: string: Advanced Organic Spinach
4: string: 0.06g

**Figure 8.** Organic vegetable information view in proposed system.

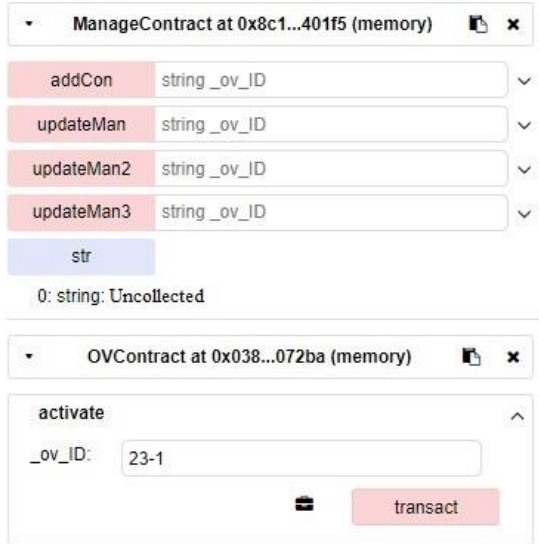

**Figure 9.** Organic vegetable transaction in proposed system.

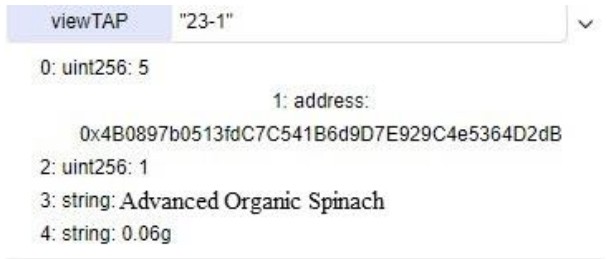

**Figure 10.** Organic vegetable transaction checks in proposed system.

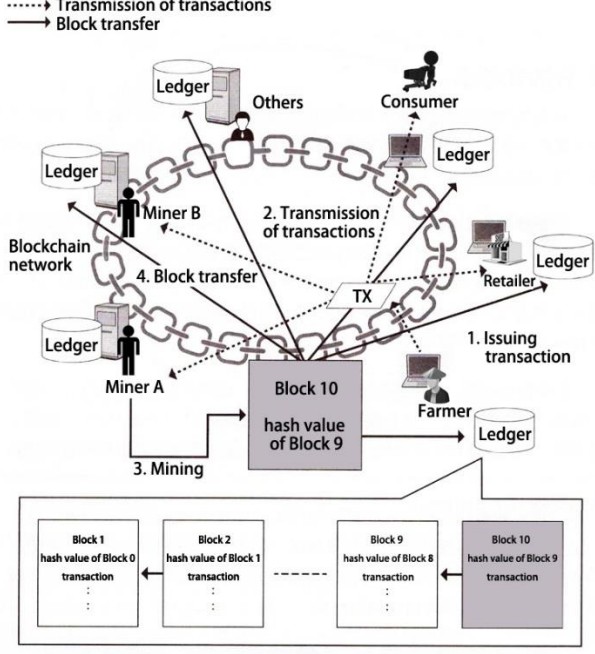

**Figure 11.** Scenario of transaction.

### 5.3. Dispute Level

When there is a dispute in a transaction, the dispute can be initiated in terms of the category. As shown in Figure 12, the organic vegetable number is 23-1. The organic vegetable number 23-1 and the reason for the dispute is entered in the "argue" section. After raising a dispute successfully, the information in the manage contract is changed to the "Controversy Phase." If there is a balance in the organic vegetable contract, the balance is fully transferred to the cultivator. All subsequent rights to the dispute are handed over to the general.

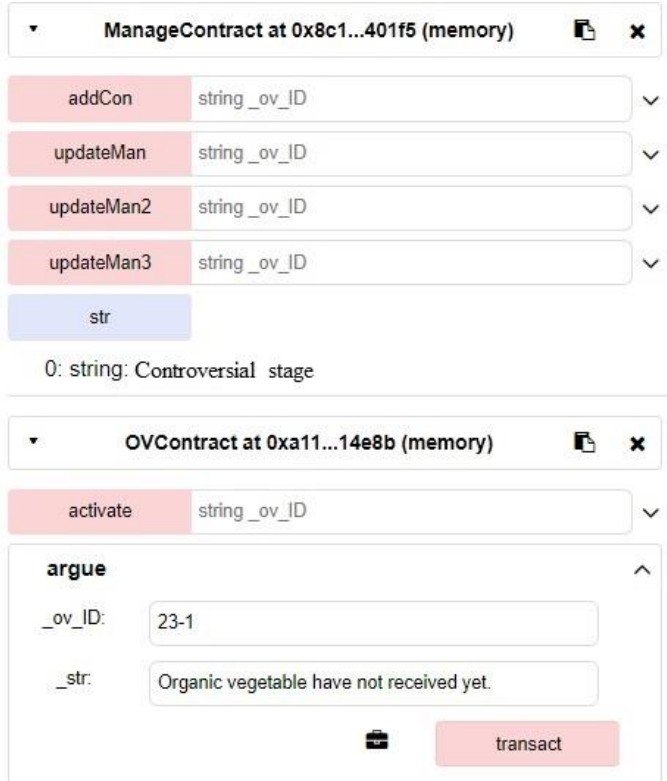

**Figure 12.** Organic vegetable argument in proposed system.

### 5.4. Security Analysis

In this section, we explain how the OVPMS (organic vegetable production and marketing) system implements the security of smart contracts.

- Reentrancy attack: the call. value() method is used to determine the balance in the contract. If the user updates the sender's balance after transmitting the balance, an attack will take effect. Note that call. value() can transmit all available gas by default, and the amount of gas transferred can be adjusted. Transfer() and send() only allow 2300 gas to be used and cannot adjust the amount of transmissions. This amount of transmissions is only sufficient to generate an event log and a transmission exception when a transmission fails. Therefore, re-entry attacks are avoided. The system uses the transfer() and send() methods, so there is no risk of re-entry attacks.
- Overflow and underflow: overflow occurs when the unit 256 type variable is greater than 256 bits. This variable changes to zero. Alternatively, when the quantity one is subtracted from a memory location containing the number 0, underflow occurs; attempts to manage underflows or overflows in Ether are very dangerous.
- Therefore, the system does not add or subtract when processing the price and has a judgment of not less than zero and not more than two while setting the price. Thus, there is no risk of a numerical overflow.

- Replay attack: in this event of repeated data transmissions, the system cannot effectively prove that this data has been received; this is referred to as a replay vulnerability. The attack is mainly conducted after the branching of the blockchain. The two parties share and trade the same data, but they do not exchange messages. This vulnerability has been added to protect against this attack since Geth 1.5.3. This study used Geth 1.6.5, thus confirming that there is no such security issue.
- Access restriction: this restriction is used for functions. For example, the access restriction feature is used only when the address of the contract is established to enforce the limits of the function. In 2017, the e-wallet parity suffered serious damage because no access restrictions were set. To prevent this vulnerability, the OVPMS system has a set access restriction while calling functions.

## 6. Conclusions

In this study, we mentioned that people have paid an increasing amount of attention to their own health in recent years and have encountered problems pertaining to food safety and the authenticity of organic vegetables. This study designed an OVPMS based on the Ethereum blockchain that may solve the problems mentioned in this paper. First, the openness and transparency features of the blockchain are used to solve the authenticity of the organic vegetable production record and test results. Thus, the price details are made available to the public and thus cannot be subject to maliciously agreed unreasonable prices. Second, the system combines the roles of the cultivator. When someone has an objection (for example, the price is unreasonable or a problem in the trading stage), the dispute can be submitted to the arbitrator for processing. Third, smart contracts are used for payment collection. Finally, our system uses the Remix IDE for testing, and the results are feasible.

**Author Contributions:** Conceptualization, D.-H.S.; Formal analysis, K.-C.L and Y.-T.S.; Investigation, K.-C.L. and Y.-T.S.; Methodology, D.-H.S.; Project administration, D.-H.S.; Validation, K.-C.L. and P.-Y.S.; Writing, review and editing, P.-Y.S.

**Funding:** This research received no external funding.

**Conflicts of Interest:** The authors declare no conflict of interest.

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
