# Peer review of "A Simulated Organic Vegetable Production and Marketing Environment by Using Ethereum"

_electronics, doi:10.3390/electronics8111341_

Round 1

Reviewer 1 Report

The authors made an excellent work, but it need some parts of the manuscript to be corrected in order to be more accurate.

There is no justification why the authors use ethereum instead other ways The 7 algorithms that have been used for authors, should be presented in one flow chart. Figures 5-9 need to be presented in one figure and as flow chart. Moreover math equations need to be used in order the algorithms in better way.

Author Response

The authors made an excellent work, but it need some parts of the manuscript to be corrected in order to be more accurate.

There is no justification why the authors use Ethereum instead other ways The 7 algorithms that have been used for authors, should be presented in one flow chart. Figures 5-9 need to be presented in one figure and as flow chart. Moreover, math equations need to be used in order the algorithms in better way.

ANS: Thank you for the reviewer’s valuable comment. In terms of blockchain technology, Ethereum ‘s resources are easy to obtain and is an open source. Blockchain’s research by using Ethereum are covered by many area (Hu et al.,2019; Pinna et al.,2019; Chen et al., 2019; Wang et al., 2019; Hu, Fan and Gao, 2019). Therefore, this study was carried out using Ethereum under the cost of the public. The authors have also published all the source code of Ethereum in https://github.com/elaine15999/solidity for demonstration. As suggested, the 7 algorithms that have been used for authors have been integrated and presented in one flow chart at Fig. yy in this new revised version. However, Figures 5-9 in original version are all screenshot of Ethereum implementation and the authors found that it is hard to shown in one figure. And, after close examine the math in Algorithms, the authors are fail to find more math representation. Thank you for the reviewer’s effort.

Reviewer 2 Report

Noted a typo "blochchain" in the title.
‘4.1 Regstered..’ This paper presents a solution and framework leveraging Ethereum based-smart contracts to prove authenticity the record of organic vegetable production and sales. The proposed system is details and aspects related to the system architecture, design, and implementation algorithms. The presented aspects and details are generic enough and can be applied to provide authenticity and transparency to any crop or produce in agricultural supply chain. For further improvements need to be considered issues regarding identity registration and data privacy of the actors involved in it.

Author Response

Noted a typo "blochchain" in the title. ‘4.1 Regstered..’ This paper presents a solution and framework leveraging Ethereum based-smart contracts to prove authenticity the record of organic vegetable production and sales. The proposed system is details and aspects related to the system architecture, design, and implementation algorithms. The presented aspects and details are generic enough and can be applied to provide authenticity and transparency to any crop or produce in agricultural supply chain. For further improvements need to be considered issues regarding identity registration and data privacy of the actors involved in it.

ANS: Thank you for the reviewer’s valuable comment. As suggested, the authors have corrected spelling errors and add more references in this new revised version. The information in the processing of all transactions will be recorded in the blockchain network. Because of the audit needed, the identity of the farmer and the seeder is open, but it only revealed its e-wallet address as shown in Table 1 in this new revised version, so the data privacy leak is not as serious as imagined. Thank you for the reviewer’s effort.

Reviewer 3 Report

This paper presents a novel system developed using Solidity in Ethereum for storing information about organic vegetable production and sales. The authors use  blockchain technology in order to guarantee the authenticity of organic vegetables.  While the paper contains many pseudo codes of the functions implemented,  it is not clear how the system works in reality.  Why should consumers update the ownership details of the organic vegetables.  How can a dispute be resolved? It would be good to illustrate the working with a case scenario of real-life transactions for production, sales and dispute.  The paper claims that the system can increase the sales of organic vegetables, and solve the problem of agricultural ecological environment pollution without any study conducted in reality!  Too many ambitious claims made without any justification.  Also there are spelling errors throughout the paper (right from the title).  These revisions are essential along with more referencing.

Author Response

This paper presents a novel system developed using Solidity in Ethereum for storing information about organic vegetable production and sales. The authors use  blockchain technology in order to guarantee the authenticity of organic vegetables.  While the paper contains many pseudo codes of the functions implemented, it is not clear how the system works in reality. 

ANS: Thank you for the reviewer’s valuable comment. This study currently focuses on the technology used in the production and marketing of organic vegetables with blockchain, so this article is only for the blockchain function. The system works in reality will be studied in the future to establish a complete system and participate in role interaction. However, the authors have published all the source code of Ethereum in https://github.com/elaine15999/solidity for demonstration. The authors have added this reply at first paragraph in section 5 in this new revised version. Thank you for the reviewer’s effort.

Why should consumers update the ownership details of the organic vegetables.

ANS: Thank you for the reviewer’s valuable comment. This study defines that consumers are not the ultimate organic vegetable owners. This study allows first-hand or even auctions. Consumers are still resold to other consumers after purchase, so consumers can update the price or ownership of organic vegetables. The authors have added this reply at section 3.1 in this new revised version. Thank you for the reviewer’s effort.

How can a dispute be resolved? It would be good to illustrate the working with a case scenario of real-life transactions for production, sales and dispute. 

ANS: Thank you for the reviewer’s valuable comment. There may be disputes during a transaction. If the farmer sells the organic vegetables to the consumer and does not receive the payment, the farmer will not be able to update the status of the smart contract, which will result in a dispute. At this point, the transaction will be stopped first and Arbitrators are arbitrated (Hasan & Salah, 2018). A case scenario of real-life transactions for production, sales and dispute are shown at last paragraph in section 3.1 in this new revised version. Thank you for the reviewer’s effort.

The paper claims that the system can increase the sales of organic vegetables, and solve the problem of agricultural ecological environment pollution without any study conducted in reality!  Too many ambitious claims made without any justification.  Also there are spelling errors throughout the paper (right from the title).  These revisions are essential along with more referencing.

ANS: Thank you for the reviewer’s valuable comment. As suggested, the authors have deleted all the ambitious claims, corrected spelling errors and add more references in this new revised version. Thank you for the reviewer’s effort.

Round 2

Reviewer 3 Report

Most of the comments have been addressed.  However, one crucial aspect to incorporate is to give a real case study flavour for the application of the proposed system in  the food industry.

The title is over-ambitious as well as the text including both production and marketing together always while in reality the production process and marketing process are quite different.

For acceptance, a case scenario should be taken and applied for all transactions  in a supply-chain.  Figures 7-11 need to have realistic data an not system oriented data used just for testing done by programmers. 

The cost analysis has no comparison with any other system.  Hence, the meaning conveyed by Table 2 provides no value to the readers.

Author Response

Most of the comments have been addressed.  However, one crucial aspect to incorporate is to give a real case study flavour for the application of the proposed system in the food industry.

ANS: Thank you for the reviewer’s valuable comment. The authors have try our best to find a real case of production and marketing company in these days. A case study from Albertsons Companies is the only citation the authors can obtain. The authors have added this case study at section 3.1 in this new revised version. Thank you for the reviewer’s effort.

The title is over-ambitious as well as the text including both production and marketing together always while in reality the production process and marketing process are quite different.

ANS: Thank you for the reviewer’s valuable comment. As suggested, the authors have corrected the title to “A Simulated Organic Vegetable Production and Marketing Environment by Using Ethereum” to make the title humbler in this new revised version. Thank you for the reviewer’s effort.

For acceptance, a case scenario should be taken and applied for all transactions  in a supply-chain.  Figures 7-11 need to have realistic data an not system oriented data used just for testing done by programmers.

ANS: Thank you for the reviewer’s valuable comment. As suggested, the authors have added a transaction scenario at section 5.2 to make it more clear in transaction representation in this section in this new revised version. Due to Ethereum is a blockchain simulated environment, the authors can only show all transaction as the authors can. Figures 7-11 are left only for validation of proof-of-work. However, it represent transaction result in Ethereum. If reviewer insist, then it can delete for final version without doubt. Thank you for the reviewer’s effort.

The cost analysis has no comparison with any other system.  Hence, the meaning conveyed by Table 2 provides no value to the readers.

ANS: Thank you for the reviewer’s valuable comment. As suggested, the authors have deleted this section in this new revised version. Thank you for the reviewer’s effort.